# Evaluating Peripheral Vision as an Input Transformation to Understand Object Detection Model Behavior

**A. Harrington**[1,2]                        ANNEKH@MIT.EDU
**V. DuTell**[1,2]                          VASHA@MIT.EDU
**M. Hamilton**[1]                         MARKTH@MIT.EDU
**A. Tewari**[1]                          AYUSHT@MIT.EDU
**S. Stent**[3]                       SIMON.STENT@TRI.GLOBAL
**W. T. Freeman**[1]                         BILLF@MIT.EDU
**R. Rosenholtz**[1,2]                        RRUTH@MIT.EDU

[1]*MIT CSAIL*
[2]*MIT Brain and Cognitive Sciences*
[3]*Toyota Research Institute*

## Abstract

Incorporating aspects of human gaze into deep neural networks (DNNs) has been used to both improve and understand the representational properties of models. We extend this work by simulating peripheral vision – a key component of human gaze – in object detection DNNs. To do so, we modify a well-tested model of human peripheral vision (the Texture Tiling Model, TTM) to transform a subset of the MS-COCO dataset to mimic the information loss from peripheral vision. This transformed dataset enables us to (1) evaluate the performance of a variety of pre-trained DNNs on object detection in the periphery, (2) train a Faster-RCNN with peripheral vision input, and (3) test trained DNNs for corruption robustness. Our results show that stimulating peripheral vision helps us understand how different DNNs perform under constrained viewing conditions. In addition, we show that one benefit of training with peripheral vision is increased robustness to geometric and high severity image corruptions, but decreased robustness to noise-like corruptions. Altogether, our work makes it easier to model human peripheral vision in DNNs to understand both the role of peripheral vision in guiding gaze behavior and the benefits of human gaze in machine learning. Data and code will be released at https://github.com/RosenholtzLab/coco-periph-gaze

**Keywords:** peripheral vision, object detection, image transformation, dataset

## 1. Introduction

Aspects of human gaze such as search patterns (Chen et al., 2021), foveation (Jonnalagadda et al., 2021) and V1-inspired filtering (Dapello et al., 2020) have been increasingly studied in deep neural networks (DNNs) to model human visual processing and improve learned representations. One component of human gaze that is less often explored is peripheral vision. Peripheral vision describes the process in which human vision represents the world with decreasing fidelity at greater eccentricities, i.e. farther from the point of fixation. Over 99% of the human visual field is represented by peripheral vision. While it is thought to be a mechanism for dealing with capacity limits from the size of the optic nerve and visual cortex, peripheral vision has also been shown to serve as a critical determinant of human performance for a wide range of visual tasks (Whitney and Levi, 2011; Rosenholtz, 2016). Additionally, the interactions between peripheral vision and foveal vision play an important

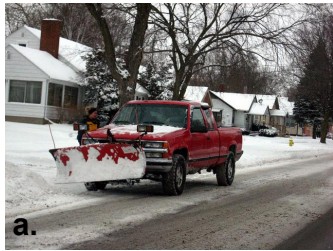 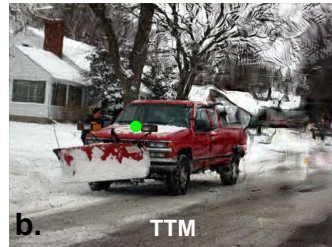 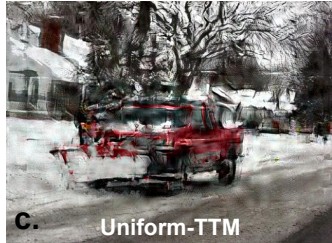

Figure 1: **The Texture Tiling Model**. **(a)** Original image. **(b)** TTM transformed image assuming the green dot is the fixation point. **(c)** TTM modified to uniformly render an image at one distance in the periphery. $15°$ is shown here.

role in directing the timing and location of saccades (Laubrock et al., 2013; Ludwig et al., 2014; Nuthmann, 2014).

Despite its importance in human gaze perception, accurately modeling peripheral vision in DNNs is challenging. Current DNN approaches are disjoint and a number of them require specialized architectures (Min et al., 2022), only model a loss of resolution (Pramod et al., 2022) – which is insufficient to predict effects of peripheral vision that are critical to gaze behavior like crowding (Balas et al., 2009)–, or rely on style transfer approaches (Deza and Konkle, 2020) which are not as well tested as statistical models. In human vision science, peripheral vision has been well-modeled with multi-scale pyramid-based image transformations that, rather than predicting visual performance on a particular task, instead output images transformed to represent the information available in peripheral vision. Humans viewing these transformed images perform visual tasks such as search with an accuracy that well predicts human performance while fixating the original images (Ehinger and Rosenholtz, 2016; Rosenholtz et al., 2012b; Freeman and Simoncelli, 2011).

In this work, we leverage one of these pyramid-based peripheral vision models, the Texture Tiling Model (TTM) (Rosenholtz et al., 2012b), to simulate peripheral vision in a variety of DNN models. We do so by modifying TTM to use a uniform rather than a foveated pooling operation (uniform-TTM); this allows us to model a single point in the periphery without having to choose a fixation. We use uniform TTM to render a popular object dataset, MS-COCO (Lin et al., 2014), to simulate peripheral vision at the input level for DNNs. Evaluating a variety of pre-trained models on the transformed images, we find that all show significant performance degradation at far distances (eccentricities) in the periphery – that is, fewer objects are confidently and correctly detected. Training with the transformed images helps models perform better under peripheral viewing conditions; additionally, we see increased robustness to certain geometric image corruptions but interestingly not noise corruptions. Overall, we demonstrate that critical aspects of peripheral vision can be modeled as an input transformation in DNNs. Doing so enables us to understand the impact of peripheral vision constraints on detection DNNs which, in future, will help us understand the role of peripheral vision in guiding human gaze behavior.

## 2. Background

### 2.1. Peripheral Vision

Foveal and peripheral vision are thought to be closely related in the context of human gaze, where information from both is integrated across saccades (Stewart et al., 2020). On its own, peripheral vision, has been successfully modeled as a loss of information in representation space (Rosenholtz et al., 2012b; Freeman and Simoncelli, 2011), where models like TTM (Rosenholtz et al., 2012b) perform a texture-processing-like computation of local summary statistics within pooling regions that grow with eccentricity and tile the visual field. This model has been tested to well predict human performance on an extensive number of behavioral tasks, including peripheral object recognition, visual search, and a variety of scene perception tasks (Ehinger and Rosenholtz, 2016). Although TTM is powerful, the computational requirements of synthesizing TTM transforms make it impractical to use online at the large scale of DNNs. Synthesizing a single TTM transform image can take 5+ hours. This has been addressed in part by (Brown et al., 2021), which modified the optimization process for transform generation with gradient descent, allowing GPU-optimization, and (Deza et al., 2017) and (Wallis et al., 2017) which incorporated style-transfer into the procedure. However, these models are not as well validated on human performance as TTM, and most are still not fast enough to use during DNN training. To facilitate large experiments, we render a dataset that pre-computes these images with a more flexible fixation scheme.

### 2.2. Human-Inspired Deep Neural Networks

Extensive work has been done in creating biologically-inspired object recognition models. A number of these models have been shown to impart representational benefits such as robustness to occlusion (Deza and Konkle, 2020) and adversarial robustness (Vuyyuru et al., 2020; Dapello et al., 2021; Guo et al., 2022). It has also been suggested that adversarial training alone can improve human perceptual alignment (Dapello et al., 2020; Feather et al., 2022; Ilyas et al., 2019; Harrington and Deza, 2022).

Despite clear benefits on recognition tasks, modeling human vision is less explored in more complex tasks like object detection. One exception to this includes FoveaBox which takes inspiration from foveation in human vision to simultaneously predict object position and boundaries without anchors (Kong et al., 2020). Search tasks have also been explored with COCO-Search18 gaze tracking data (Chen et al., 2021), which was used to build an inverse reinforcement learning model that out-performed standard baselines (Yang et al., 2020). Additionally, training on a stylized version of COCO (Michaelis et al., 2019) (much like the stylized ImageNet work which reduced texture bias and increased shape bias in recognition models (Geirhos et al., 2018)) was shown to increase corruption robustness in object detection DNNs. Peripheral vision, however, is thought to use texture-like representations, and is critically involved in tasks where context matters like detection. Testing peripheral vision in tasks like detection is key to understanding the benefits and trade-offs of modeling human gaze in DNNs.

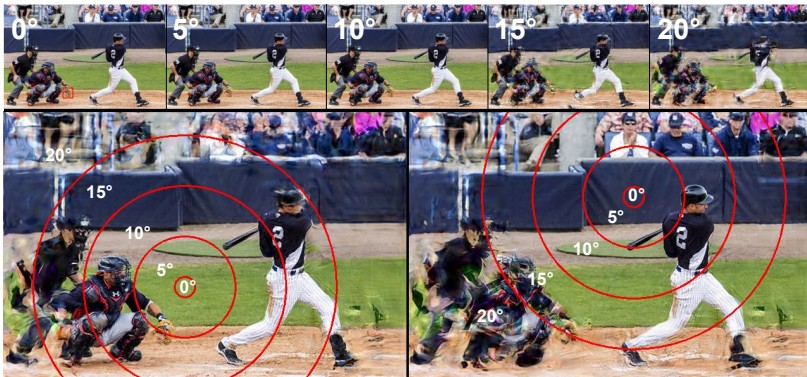

Figure 2: **Pseudo-Foveated images are created by stitching together Uniform TTM images.** Pseudo-foveation allows us to simulate a fixation at any point in an image. The top row shows TTM transforms for a single eccentricity, and the bottom row shows two pseudo-foveated TTM transforms for different fixations.

## 3. Peripheral Vision Dataset

To model critical aspects of peripheral vision without having to render a large number of fixations, we use a modified version of TTM that relies on uniform, rather than foveated pooling. In original TTM, pooling regions grow linearly with eccentricity, but for uniform TTM, we fix the pooling region size to match the ring of pooling regions at a single eccentricity (see Appendix Sec. 8.1). For example, we set the pooling region size to correspond to $15°$ eccentricity, as in Figure 1. We refer to images transformed by uniform-TTM as peripheral transform images. With uniform pooling, we can create images that show the information available as if each pooling region appeared at the same eccentricity. Though this represents an impossible situation, it provides two practical advantages: (1) the ability to shift the modeled fixation by stitching together pre-computed, uniformly transformed images to create pseudo-foveated images in only 50ms (see Appendix Sec. 8.2 and Fig. 2), and (2) to evaluate both human and machine performance for an entire image at a single eccentricity.

We apply Uniform TTM to the COCO dataset, creating a dataset which contains images transformed like peripheral vision. This transformed dataset allows us to use the highly tested Texture Tiling as an input pre-processing step to train and evaluate DNNs. In our work, we render images that capture the amount of information available to humans at $(5°, 10°, 15°, 20°)$ in the periphery, assuming 16 pixels per degree. (For reference, the width of a full-screen image on a laptop at a typical viewing distance subtends $20° - 40°$). We render the entire validation set of COCO to stimulate peripheral vision, along with $55,000$ images from the training set. With this dataset, we can evaluate what objects a DNN can detect in the periphery.

## 4. Evaluating Baseline Model Performance

Measuring object detection performance on peripheral-transformed COCO using the original COCO ground truth labels, we see in Table 1 for a variety of pre-trained models that

| Model Arch | AP 0° | AP 5° | AP 10° | AP 15° | AP 20° |
|---|---|---|---|---|---|
| DINO-FocalNet-Large (Zhang et al., 2022) | **58.4** | **51.6** | **44.4** | **20.2** | **15.0** |
| DINO-Swin-Tiny (Zhang et al., 2022) | 51.3 | 44.0 | 34.1 | 11.1 | 7.6 |
| Detr-R50 (Carion et al., 2020) | 42.0 | 35.2 | 25.1 | 6.9 | 4.5 |
| RetinaNet-R50 (Lin et al., 2017) | 38.7 | 31.5 | 22.1 | 6.9 | 5.0 |
| Faster-RCNN-X101 (Ren et al., 2015) | 39.6 | 32.6 | 21.8 | 6.6 | 4.7 |
| Faster-RCNN-R50 (Ren et al., 2015) | 36.7 | 29.4 | 19.9 | 5.9 | 4.2 |
| All ° Train Faster-RCNN-R50 | 33.8 | 30.5 | 28.1 | 15.8 | 12.7 |
| All ° FT Faster-RCNN-R50 | 36.1 | 31.8 | 27.7 | 13.9 | 10.8 |

Table 1: **Average Precision (AP) on the validation set COCO and peripheral-transformed COCO**. 0° refers to performance on unchanged MS-COCO data. The other AP values correspond to different eccentricities of peripheral transform images.

average precision (AP) degrades with distance in the periphery. De-noising models, which have the highest baseline scores, perform the best at peripheral object detection overall compared to the other architectures measured. Performance likely degrades because the uniform-TTM transform is potentially out of distribution for models, and at farther eccentricities, objects have a greater potential to move due to the larger pooling regions used.

## 5. Training with Peripheral Vision Input

Given the drop-off in performance for baseline models when processing peripheral visual input, we wondered if object-detection models fine-tuned or trained with peripheral-transform images could achieve improved performance. To that end, we fine-tune and train from scratch a ResNet-50 backbone Faster-RCNN-FPN detection model with uniform TTM images from COCO (See Appendix Sec. 8.3).

For both fine-tuned and models trained from scratch, we see a marked performance improvement as compared to the baseline RCNN-R50 model (Table 1, bottom), with the scratch trained model outperforming the fine-tuned trained model at higher eccentricities, and the fine-tuned model outperforming the scratch trained only for original images and the lowest eccentricity (5°). This quantitative improvement is validated qualitatively with more reasonable and stable predictions, as well as tighter bounding boxes (Figure 3). For fine-tuning (plotted as All° FT RCNN) we start from detectron2's (Wu et al., 2019) pre-trained model and use a 3x iteration scheme with a lowered learning rate. When training from scratch (plotted as All° FT RCNN), we use the default 3x training configuration in detectron2.

While it is clear that improved performance can be attributed to in-distribution training data for the RCNN model, we note that no trained model out-performed the baseline DINO-FocalNet model. Future work should aim to tease out whether architecture, de-noising training schemes, or other aspects of this model contribute to its strong performance on peripheral transform images.

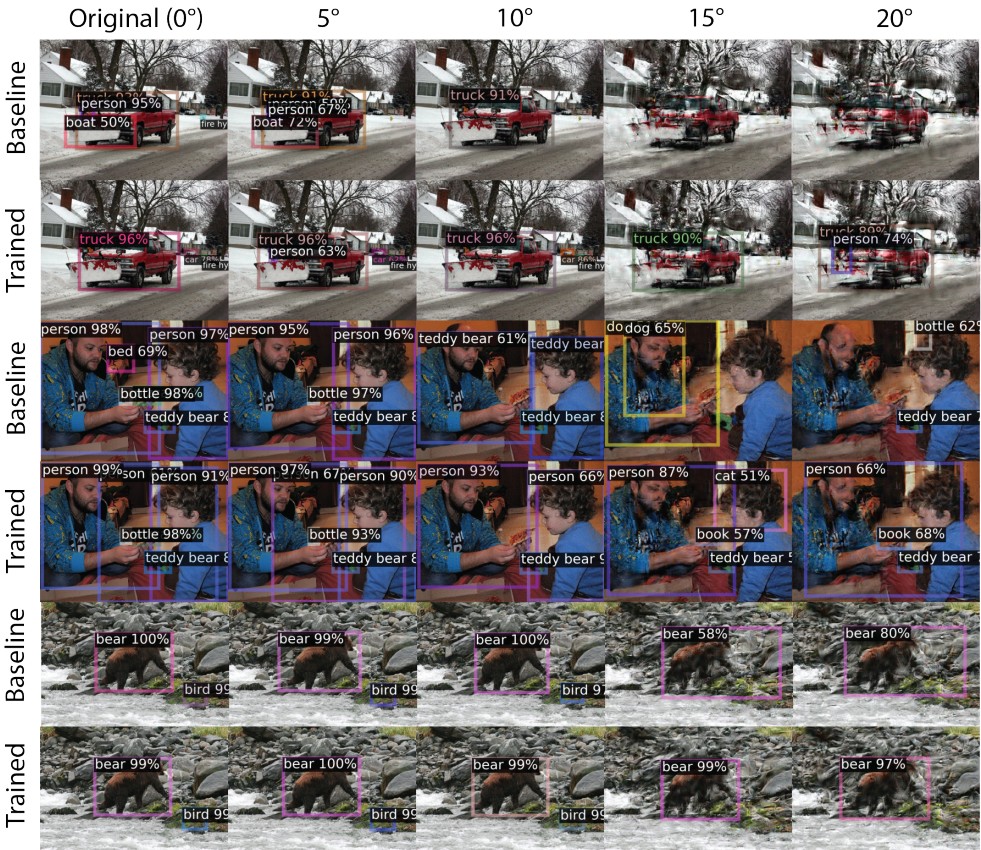

Figure 3: **Training improves object detection on Uniform TTM images.** Faster-RCNN R50 FPN trained from scratch on peripherally transformed images retains more stable predictions with more accurate bounding boxes as eccentricity increases when compared to the baseline Faster-RCNN R50 FPN model.

## 5.1. Corruption Robustness

| Model | mAP$_c$ | severity= 1 | severity= 5 | brightness | elastic transform |
|---|---|---|---|---|---|
| Faster-RCNN | 17.34 | **25.08** | 9.57 | **28.29** | 11.94 |
| All ° FineTune | **17.47** | 24.80 | 9.84 | 27.31 | 14.48 |
| All ° Train | 16.72 | 23.28 | **9.88** | 25.31 | **15.02** |

Table 2: **Corruption Robustness Average Precision on COCO Val**. All models are Faster-RCNN ResNet50 FPN (Ren et al., 2015). (mAP$_c$) average precision on COCO-C (Michaelis et al., 2019). Columns (severity= 1) and (severity= 5) report AP over all corruptions at a single severity level. Last two columns report AP for the worst and best performing corruption compared to baseline.

In order to see if there were benefits from training on peripheral-transform images beyond getting more human-like detection behavior, we evaluated trained models for cor-

ruption robustness using the COCO-C suite of corruptions (Michaelis et al., 2019). We find that corruption robustness improves most noticeably for geometric transformations like 'glass blur' and 'elastic transform' (See Table 2). Interestingly, performance is slightly lower than baseline for noise-like corruptions and ones that change contrast (See Fig. 4). Averaged over all corruptions, average precision is slightly higher for the model trained from scratch on peripheral-transform images than baseline.

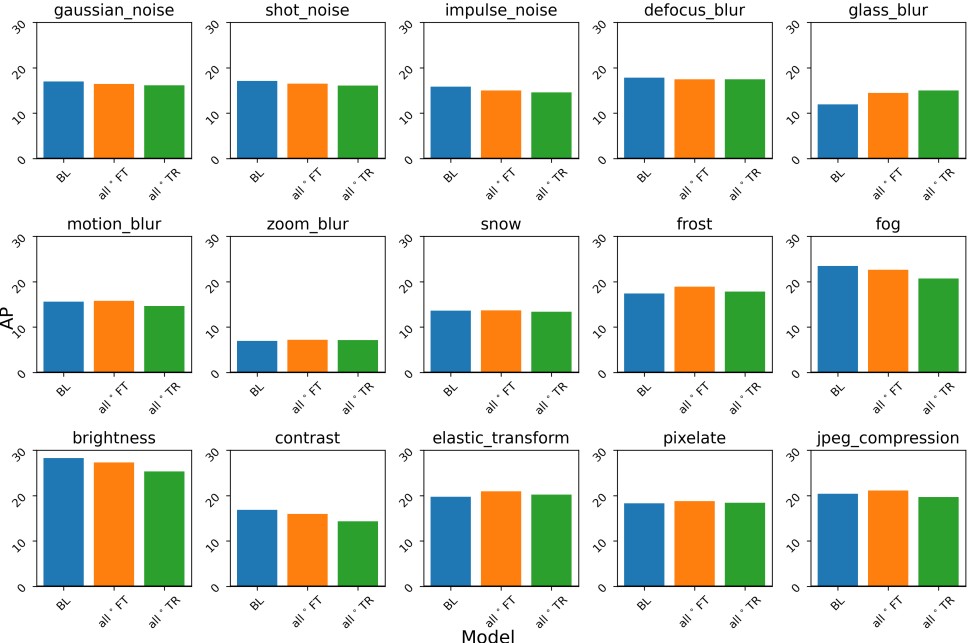

Figure 4: **Corruption Robustness of Fine-Tuned and Trained models.** We report AP for all COCO-C corruptions (Michaelis et al., 2019). BL refers to the pre-trained Faster RCNN R50 model (blue). The FT model is one fine-tuned on peripheral transform images (orange). The TR model is trained from scratch(green).

## 5.2. Generalization to Unseen Eccentricities

In order to evaluate the contribution of individual eccentricities to the performance boosts seen in the trained and fine-tuned models, we fine-tuned the RCNN model for all eccentricities separately (Appendix Fig. 6). As expected, we find that the model fine-tuned at all eccentricities performs well overall. Interestingly however, we find that models trained at only one eccentricity see marked performance boosts at other untrained eccentricities. This generalization to data outside those seen during training indicates that improvement in performance at a single eccentricity cannot be attributed only to training data at that eccentricity.

## 5.3. Effect of Object Size

In humans, peripheral vision tends to more severely degrade the perception of small objects (especially in cluttered scenes) compared to large objects, all else being equal (e.g. when

viewed at the same eccentricity). To determine the effect of TTM transform training on object detection for objects of varied sizes, we perform the same evaluation of our fine-tuned models, measuring AP-small and AP-large separately (Appendix Fig. 7 and 8). We find that fine-tuning with TTM transformed images has a much stronger advantage in improving performance for large objects, leading to an over 15% improvement in AP-large at eccentricities 10 degrees and above. This effect is most profound for large eccentricity TTM transforms, which can be extremely distorted. In contrast, AP-small shows more modest improvements at mid-range eccentricities, and little ability to perform object detection at 15 and 20 degrees. We hypothesize this is because, at these larger eccentricities, small objects are so distorted that detection is nearly impossible – even for humans.

## 6. Discussion

In our work, we explore how a human-gaze-inspired input transformation affects DNN performance on object detection. Specifically, we simulate peripheral vision in object detection DNNs by creating a dataset of images transformed to mimic peripheral vision. Looking at model performance in the periphery, we see that a variety of model architectures struggle to detect objects in the far periphery. Training on peripheral transform images improves performance on peripheral detection, but even then, we do not achieve baseline-line level performance. This suggests that training data alone is not sufficient for models to detect objects under peripheral viewing conditions. Specialized architectures or de-noising models like DINO models might be necessary to perform well at the peripheral viewing task.

One limitation of our work is that many of our evaluations calculate average precision using baseline-COCO annotations. The TTM transformation, however, inherently has the potential to change the size and location of objects – especially when modeling farther eccentricities. It is possible, that some of the degradation in performance we see on far periphery images may partially be attributable those changes introduced by TTM, but we also observe that performance degrades significantly on peripheral-transform images – even for models trained on the TTM-transformation. As a result, we expect that the degradation in performance we see is not explainable by the annotations alone. In future, creating a more human-inspired detection test to evaluate models will be important for fully understanding the effect of the TTM transform on models.

Another result from our work is that training on peripheral transform images increases robustness to geometric corruptions, but decreases robustness to noise corruptions. Although human visual representations may contribute to human robustness to adversarial noise Ilyas et al. (2019); Dapello et al. (2020) – especially the texture-like representations of peripheral vision (Harrington and Deza, 2022) – the TTM-transform itself more closely resembles geometric corruptions and this is evident in our robustness evaluations. While we do not evaluate the adversarial robustness of our trained models, it appears that more work is needed to fully understand the relationship between peripheral vision and robustness.

Overall, our work in modifying TTM and applying it to detection makes it easier to model peripheral vision in DNNs – which is critical to understanding the impact of human gaze on machine learning models. Our transformation allows us to evaluate DNNs using key aspects of peripheral vision to helps us understand how DNNs respond to human-gaze constraints. This will enable us to build models with more human-like characteristics and to explain model behavior in the context of human gaze.

## 7. Acknowledgements

This work was funded by the Toyota Research Institute, CSAIL MEnTorEd Opportunities in Research (METEOR) Fellowship, US National Science Foundation under grant number 1955219, and National Science Foundation Grant BCS-1826757 to PI Rosenholtz. The authors acknowledge the MIT SuperCloud (Reuther et al., 2018) and Lincoln Laboratory Supercomputing Center for providing HPC resources that have contributed to the research results reported within this paper.

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

## 8. Appendix

### 8.1. Uniform TTM

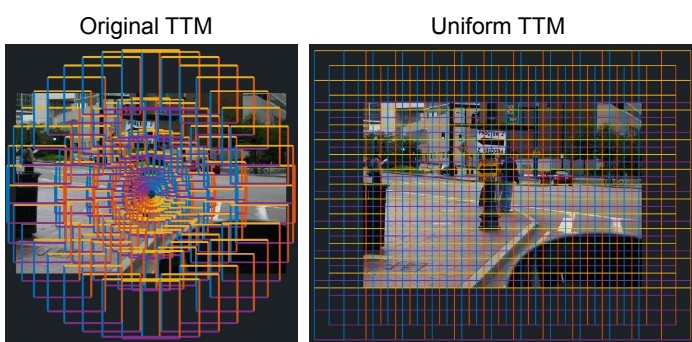

Figure 5: **Pooling Regions in Original and Uniform Texture Tiling Models (TTM).** Original TTM (Rosenholtz et al., 2012a) is foveated, so its pooling regions are small around the fixation point and grow farther from the fixation. We adapt TTM to use a fixed pooling region size everywhere in the image (Uniform TTM). The size is determined by the distance in the periphery being modeled.

### 8.2. Pseudo-Foveation

Using the uniform TTM transforms, one can quickly simulate a fixation anywhere in an image, essentially generating a foveated TTM transform at low computational cost. We call the process pseudo-foveation. To create pseudo-foveated images, we stitch together uniform TTM transforms rendered at multiple eccentricities (Figure 2). For the fovea, we insert a circle crop of the original image. Then we add cropped rings from the uniform TTM transforms, centering each ring at the eccentricity it was rendered at ($5°$ degree TTM transform is centered at $5°$ eccentricity in the image). To reduce edge effects, we linearly blend the border between uniform TTM transform crops; all borders are weighted equally in the blending. With pre-computed transforms, this process of pseudo-foveation takes only 50ms per image, more efficient at generating images than foveated TTM, making it feasible to incorporate into the dataloading loop for training and testing DNNs.

### 8.3. Training Procedure

We fine-tuned and train from scratch the Faster R-CNN model from the Detectron2 library (Wu et al., 2019) (faster_rcnn_r50_fpn) using a mixture of original training images, and TTM transforms for varying eccentricities. We use 55,000 images from each eccentricity along with original COCO images. Fine-tuning was trained for 180,000 iterations starting from the weights of a pre-trained R-CNN from (Wu et al., 2019). We set the solver to step at 120,000 and 160,000. We set the base learning rate to $3 \times 10^{-4}$. All other training parameters are the same R-CNN training parameters in (Wu et al., 2019) as the baseline model. To train from scratch, we use the same 3x training schedule provided in (Wu et al., 2019) for Faster RCNN R50 FPN models (starting from an ImageNet trained ResNet50 backbone, training for 270,000 iters, 16 images per batch).

### 8.4. Fine-Tuning on Individual Eccentricities

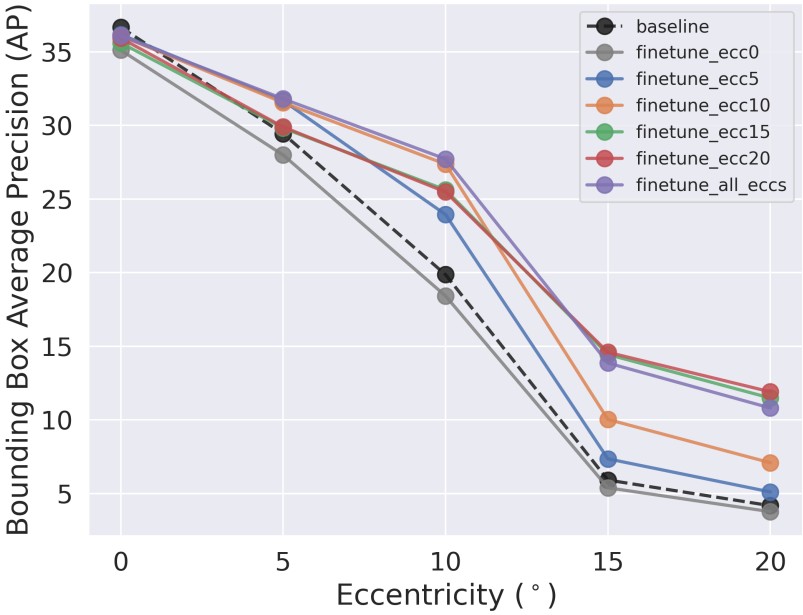

Figure 6: **Faster R-CNN R50 Object Detection Bounding Box AP.** We fine-tune an R-CNN on peripheral transform images and compare their AP to the baseline model. Fine-tuning does not improve performance on original COCO images (0°), but does improve performance at all eccentricities. Interestingly, fine-tuning on individual eccentricities generalizes to other eccentricities, resulting in performance boosts at untrained eccentricities.

## 8.5. AP Small and Large

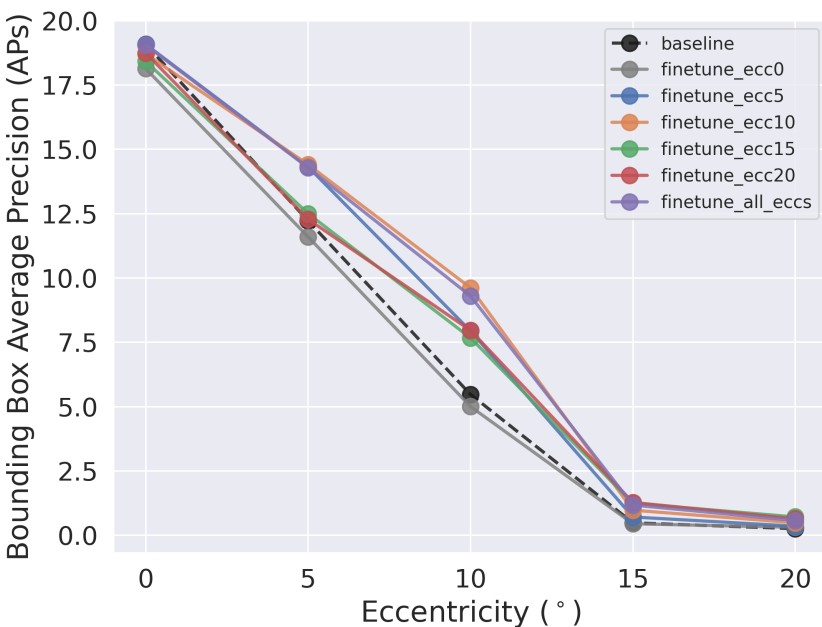

Figure 7: **Faster R-CNN R50 Object Detection Bounding Box AP for Small Objects.** We fine-tune an R-CNN on peripheral transform images and compare their AP small to the baseline model. All models generally do poorly on smaller objects. Fine-tuning does not improve performance on original COCO images, but it does slightly improve performance on small eccentricity peripheral transform images like 5 and 10°.

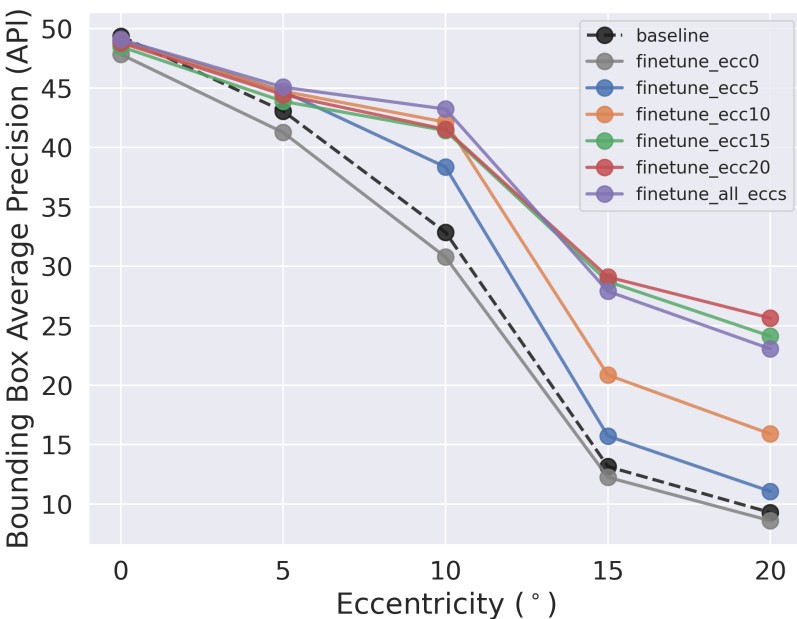

Figure 8: **Faster R-CNN R50 Object Detection Bounding Box AP for Large Objects.** All models generally do better at large objects than small objects, and fine-tuning on uniform transform images improves large object performance on peripheral transform images much more than small objects at farther eccentricities.

