# OpenReview forum: "Evaluating Peripheral Vision as an Input Transformation to Understand Object Detection Model Behavior"
_NeurIPS.cc/2023/Workshop/Gaze_Meets_ML — Gaze Meets ML 2023 Poster_

### Official Review · Reviewer_2uQQ · 2023-10-14
**Studies on the effect of TTM peripheral vision on object detectors**

**Rating:** 6
**Confidence:** 3

**Review:**

The authors modify the Texture Tiling Model, TTM approach for real-time peripheral vision simulation. They generate TTMs using uniform instead of foveated pooling, which leads to TTM images that are independent of the fixation point. To approximate fixation point dependent TTM images they stich together pre-computed uniform TTM images. This procedure is considerably faster than generating and applying the non-uniform TTM directly.

They utilize their efficient TTM approach to improve and quantify the detection performance of state-of-the-art object detectors on peripheral object detection. Additionally, they evaluate if training on uniform TTM-transformed images increases the robustness of neural networks to image corruptions.

From the conducted experiment I see no clear benefit from training models with uniform TTM-transformed images beyond robustness increases for directly related corruptions. However, the approximation of TTM using pre-computed uniform TTM transformation is interesting.

---

### Official Review · Reviewer_zvAH · 2023-10-22
**Unsurprising, but thorough evaluation of the impact of peripheral eccentricity on object detection models (e.g., Faster-RCNN)**

**Rating:** 7
**Confidence:** 3

**Review:**

The authors present a solid set of experiments on peripheral vision by adapting Texture Tiling Model (TTM) by Rosenholtz et al., 2012. They create uniformly transformed images at 5, 10, 15, and 20 degrees of eccentricity. Based on these datasets, they perform experiments on (pretrained) object detection models, Faster-RCNN models fine-tuned with eccentric images, and Faster-RCNN models trained from scratch with eccentric images. In general, results are largely unsurprising, but the authors included a fair and detailed discussion of the limitations of their study.

Strengths:
* Present an efficient algorithm based on TTM for creating arbitrary foveated images (around 50ms per image, compared to >5h).
* Clearly evaluate performance drop when using transformed images at increased eccentricities, validating that object detection is indeed more difficult in the periphery.
* Perform experiments on small and large objects separately, showing that large object performance benefits from fine-tuning at high eccentricities, and small objects are poorly distinguishable at high eccentricities.

Weaknesses:
* As mentioned by the authors themselves, the corruptions on which the trained models perform well align with the nature of the TTM, and thus the results are non-surprising and provide little insight into human vision.
* The authors present a way to generate foveated images from the uniformly transformed images, but it remains unclear if and where these are used in the paper. In addition, this algorithm also is not tested against human performance.
* Even models trained with eccentric images remain less performant compared to the best baseline DINO model.
* Unclear whether the authors intend to share their generated TTM dataset at 5, 10, 15, and 20 degrees of eccentricity.

---

### Official Review · Reviewer_NNVH · 2023-10-23
**Evaluating Peripheral Vision as an Input Transformation to Understand Object Detection Model Behavior**

**Rating:** 7
**Confidence:** 3

**Review:**

This paper proposed to use uniform TTM to simulate the peripheral vision data and experiments on the simulated peripheral data demonstrated that exisiting object detection models performance degrdated with the increase of the ecentricities. The authors proposed to use the simulated data to retrain a resnet50-Faster RCNN model to improve the pheripheral vision, and the work shows that training on peripheral transform images improves performance on peripheral detection, but even then, the accuracy cannot out-performed the baseline DINOFocalNet model. Also the author noticed that training on peripheral transform images increases robustness to geometric corruptions, but decreases robustness to noise corruptions. Overall, the paper is well-written, the contribution of how the pheripheral vision can be simulated is convinsible.

---

### Meta-Review · Area_Chair_zwtC · 2023-10-26

**Recommendation:** Accept (Poster)
**Confidence:** 4

**Metareview:**

This paper explores integrating human peripheral vision into object detection deep neural networks (DNNs) using a modified Texture Tiling Model (TTM). It improves DNN understanding and performance under constrained viewing conditions, with increased robustness to some image corruptions but decreased robustness to noise-like corruptions.

Based on the conducted experiment, it's not evident that training models with uniformly TTM-transformed images offer significant advantages, except for enhancing robustness against directly related corruptions. The observed model performance aligns with the TTM's inherent characteristics, resulting in somewhat unsurprising results that offer limited insights into human vision. This aspect should be better discussed in the paper. Additionally, it's unclear whether the authors intend to make their generated TTM dataset available at different eccentricities, such as 5, 10, 15, and 20 degrees, which would be valuable information for the research community.

---

### Decision · Program_Chairs · 2023-10-26

Accept (Poster)